# Retrieval of Aerosol Microphysical Properties from Multi-Wavelength Mie–Raman Lidar Using Maximum Likelihood Estimation: Algorithm, Performance, and Application

**Yuyang Chang [1],\*** , **Qiaoyun Hu [1]** , **Philippe Goloub [1]** , **Igor Veselovskii [2]** and **Thierry Podvin [1]**

1   UMR 8518–LOA–Laboratoire d'Optique Atmosphérique, University of Lille, CNRS, 59650 Villeneuve d'Ascq, France
2   Prokhorov General Physics Institute, Russian Academy of Sciences, Moscow 119991, Russia
\*   Correspondence: yuyang.chang@univ-lille.fr

**Abstract:** Lidar plays an essential role in monitoring the vertical variation of atmospheric aerosols. However, due to the limited information that lidar measurements provide, ill-posedness still remains a big challenge in quantitative lidar remote sensing. In this study, we describe the Basic algOrithm for REtrieval of Aerosol with Lidar (BOREAL), which is based on maximum likelihood estimation (MLE), and retrieve aerosol microphysical properties from extinction and backscattering measurements of multi-wavelength Mie–Raman lidar systems. The algorithm utilizes different types of a priori constraints to better constrain the solution space and suppress the influence of the ill-posedness. Sensitivity test demonstrates that BOREAL could retrieve particle volume size distribution (VSD), total volume concentration ($V_t$), effective radius ($R_{eff}$), and complex refractive index (CRI = $n - ik$) of simulated aerosol models with satisfying accuracy. The application of the algorithm to real aerosol events measured by LIlle Lidar AtmosphereS (LILAS) shows it is able to realize fast and reliable retrievals of different aerosol scenarios (dust, aged-transported smoke, and urban aerosols) with almost uniform and simple pre-settings. Furthermore, the algorithmic principle allows BOREAL to incorporate measurements with different and non-linearly related errors to the retrieved parameters, which makes it a flexible and generalized algorithm for lidar retrieval.

**Keywords:** maximum likelihood estimation; retrieval of height-resolved aerosol microphysical properties; analysis of lidar measurements

## 1. Introduction

Atmospheric aerosols play a significant role in the Earth's climate change and radiative forcing. They can not only change the scattering and absorption of incident solar irradiance, but also affect the formation and optical properties of clouds through aerosol-cloud interactions [1]. The tempo–spatial variation of aerosols properties and sources makes up a dominant source of uncertainty for the assessment of the Earth's radiative forcing and the global temperature projection, although important progress has been made since the last decade thanks to the progress in both atmospheric modeling and observation systems [2].

Remote sensing is an effective way to continuously monitor the temporal and spatial distributions of aerosols and access their microphysical properties, such as particle volume size distribution (VSD), complex refractive index (CRI = $n - ik$), morphologic parameters, and so on. Passive remote sensing, like regional global surface networks [3] or space-borne instruments [4,5], is capable of providing long-term aerosol monitoring with global coverage, whereas it cannot derive height-resolved aerosol properties, which are important for accurately assessing aerosol radiative forcing [6]. In this context, light detection and ranging (lidar) technology has been widely used for atmospheric remote sensing [7].

However, at the early stage of lidar remote sensing, due to relatively high measurement uncertainty [8], the retrieval of aerosol microphysical properties was usually conducted under the assumption of the Junge VSD or known CRI, which limits the application to real measurements [9–11].

More general retrieval methods were proposed since the 1990s when the technique of Raman lidar [12,13] or High Spectral Resolution Lidar (HSRL) [14], which is capable of simultaneously measuring the extinction coefficient ($\alpha$) and backscattering coefficient ($\beta$) with enough accuracy, was developed. Based on such an instrumental leap, on the one hand, lidar networks on a continental scale—such as the European Aerosol Research Lidar Network under the framework of the Aerosol, Clouds, and Trace gases Research InfraStructure (EARLINET/ACTRIS), Micro Pulse Lidar Network (MPLNET), and Asian Dust Lidar Network (ADNET)—have been established since 2000 to extend the spatial coverage of ground-based lidar observations [15–17]. On the other hand, a number of algorithms aimed at retrieving tropospheric aerosols from lidar measurements have been proposed. For example, a well-known method is to linearly inverse the Fredholm integral composed of VSD and optical kernels for a series of CRIs and certain radius ranges using regularization or principal component analysis. Then, a family of solutions which minimize the so-called *discrepancy* will be selected and averaged [18–23]. Another method is based on Look up Tables (LUTs), such as the arrange and average method [24], which utilizes combined measurements of $\alpha$, $\beta$, and lidar ratio (LR) at several wavelengths.

Previous studies demonstrated that "$3\beta + 2\alpha$", i.e., backscattering coefficients at 355 nm, 532 nm, and 1064 nm and extinction coefficients at 355 nm and 532 nm, is the least lidar measurements to retrieve aerosol microphysical properties [20,25–27]. However, this inversion system is ill-posed because, on the one hand, the measurements are highly interdependent on each other and, on the other hand, the number of retrieval parameters in which we are interested is usually more than the number of the measurements. Chemyakin et al. [28] pointed out that the main difficulty in lidar inversion is the non-uniqueness of the solution. Indeed, such difficulty is faced by some retrieval algorithms, such as the linear regularization method and principal component analysis method, which have to identify the proper solution space from all the "solutions" derived by performing linear inversion at every point in the searching domain composed of all non-linear parameters (e.g., CRI) [18–23]. For example, the minimum discrepancy method [20] could find two "qualified" solutions corresponding to different local minima far from each other. In this circumstance, additional constraints on the searching domain must be applied [29]. However, with the development of more advanced lidar systems, as well as the increasing need of synergy with other types of instruments, more aerosol microphysical properties non-linearly coupled with each other are expected to be retrieved quantitively. As a result, traditional linear retrieval algorithms will suffer from both increase of computational burden and algorithmic complexity. In this regard, we propose a non-linear retrieval algorithm, BOREAL (Basic algOrithm for REtrieval of Aerosol with Lidar), based on maximum likelihood estimation (MLE) to reduce the ill-posedness of $3\beta + 2\alpha$ and improve the identification of solution space by incorporating a priori constraints from multi sources. Although the statistical optimization strategy used in BOREAL allows flexibility to incorporate different types of measurements, for example, profiles of depolarization ratio ($\delta$) and fluorescence, only $3\beta + 2\alpha$ data are inverted at this preliminary stage. This study will contribute to the development of an automated aerosol retrieval of LIlle Lidar AtmosphereS (LILAS), operated under the frame of ACTRIS/EARLINET [30,31], and other LILAS-like lidar systems.

The following sections of this paper are organized as follows: in Section 2, we demonstrate the principle and implementation of the BOREAL algorithm; in Section 3, we test the algorithm through sensitivity study using synthetic data; in Section 4, to further evaluate BOREAL's performance, it is applied to a set of real aerosol events (dust, aged smoke, and urban aerosols) detected by LILAS during the SHADOW-2 campaign and in operation at the Atmospheric Observatory of LilLe (ATOLL); and Section 5 concludes this paper.

## 2. BOREAL Algorithm

### 2.1. Modeling the Problem

The optical data consisting of extinction and backscattering coefficients measured by lidars can be modeled through particle bulk single-scattering properties:

$$\alpha_\lambda = \int_{\ln r_{min}}^{\ln r_{max}} \frac{3\sigma_{ext}(\lambda, n, k, \ln r)}{4\pi r^3} v(\ln r) d \ln r + \varepsilon_{\alpha_\lambda} \tag{1}$$

$$\beta_\lambda = \int_{\ln r_{min}}^{\ln r_{max}} \frac{3\sigma_{bac}(\lambda, n, k, \ln r)}{4\pi r^3} v(\ln r) d \ln r + \varepsilon_{\beta_\lambda} \tag{2}$$

where $\sigma_{ext}$ and $\sigma_{bac}$ are extinction and backscattering cross sections of a single spherical particle, respectively, functions of wavelength $\lambda$, real part of the CRI ($n$), imaginary part of the CRI ($k$), and particle radius ($r$). The particle VSD, $v(\ln r)$, is a continuous function of $\ln r$, and $\varepsilon_{...}$ stands for the error in extinction or backscattering measurements. $\sigma_{ext}$ and $\sigma_{bac}$ can be calculated from various scattering models, such as Mishchenko et al. [32] and Yang and Liou [33].

Because of the finite number of measurements, $v(\ln r)$ is approximated by the linear combination of a set of base functions $\{\phi_j(\ln r)\}$:

$$v(\ln r) \approx \sum_{j=1}^{N} v_j \phi_j(\ln r) \tag{3}$$

where $v_j$ is the weight factor of $\phi_j$. A smooth function with continuous second derivative can be approximated by a piecewise cubic polynomial, which can be expressed as a linear combination of a B-spline basis [34,35]. On the basis of previous studies and for the sake of simplifying the computation [18,20,36,37], we utilize the B-splines of the first degree as the base functions which have the following definition:

$$\phi_j(\ln r) = \begin{cases} 0, & \ln r \leq \ln r_{j-1} \\ \frac{\ln r - \ln r_{j-1}}{\ln r_j - \ln r_{j-1}}, & \ln r_{j-1} < \ln r \leq \ln r_j \\ \frac{\ln r_{j+1} - \ln r}{\ln r_{j+1} - \ln r_j}, & \ln r_j < \ln r \leq \ln r_{j+1} \\ 0, & \ln r > \ln r_{j+1} \end{cases} \tag{4}$$

where the piecewise nodal grids are logarithmic equidistant and $r_1 = r_{min}, r_N = r_{max}$. Correspondingly, $v_j$ is equal to $v(\ln r_j)$. With the increase of the number of B-spline functions, i.e., the increase of $N$ in Equation (3), both approximation accuracy and ill-posedness will increase. To balance the two competing factors, $N$ is set to 8 in this study. We found this to be the smallest value to represent aerosol bimodal size distributions with acceptable accuracy and, at the same time, not to cause too large ill-posedness in the inversion procedure. $N = 8$ was also adopted by other studies where linear inversion methods were used for the $3\beta + 2\alpha$ data [18,21,25,38]. With Equations (3) and (4), Equations (1) and (2) can be written in the vector–matrix form:

$$\mathbf{y}_1 = \mathbf{f}_1(\mathbf{x}) + \varepsilon_1 = \mathbf{K}(n,k)\mathbf{v} + \varepsilon_1 \tag{5}$$

where $\mathbf{y}_1$ is the vector of lidar measurements. For a typical aerosol lidar with a Nd: YAG laser, $\mathbf{y}_1 = [\alpha_{355}, \alpha_{532}, \beta_{355}, \beta_{532}, \beta_{1064}]^T$. $\varepsilon_1$ represents the vector of measurement errors, $\mathbf{v} = [v_1, v_2, \cdots, v_N]^T$ collects the weight factors, and $\mathbf{x} = [\mathbf{v}^T, n, k]^T$. $\mathbf{K}$ is the kernel matrix with the elements

$$\{\mathbf{K}(n,k)\}_{ij} = \int_{\ln r_{j-1}}^{\ln r_{j+1}} \frac{3\sigma_i(n, k, \ln r)}{4\pi r^3} \phi_j(\ln r) d \ln r \tag{6}$$

where $i$ corresponds to the element of **y**. At current stage, we use the database of Dubovik et al. [37], where the kernel matrices of spherical particles and spheroidal particles with a fixed axis ratio distribution were precalculated, for fast calculation of $\alpha$ and $\beta$. Other scattering models for specific non-spherical particles, such as the super-spheroid model and the advanced bulk optical model [39,40], will be implemented in the next step.

According to the definition by Hadamard [41], Equation (5) is ill-posed, as there are typically 5 lidar measurements, but 10 parameters to be retrieved. Since most of realistic size distributions of aerosol particles are smooth functions (with continuous second derivatives), we introduce the following smoothing constraint on VSD:

$$\mathbf{y}_2 = 0 = \mathbf{f}_2(\mathbf{x}) + \varepsilon_2 = \mathbf{H}\mathbf{v} + \varepsilon_2 \tag{7}$$

where **H** is the operator to calculate the second-order difference of **v**. $\varepsilon_2$ acts as a weight factor of the constraint.

To further reduce the ill-posedness, a priori constraints are also applied to the real and imaginary parts of the CRI [42]:

$$\mathbf{y}_3 = n_\mathrm{a} = \mathbf{f}_3(\mathbf{x}) + \varepsilon_3 = n + \varepsilon_{n_\mathrm{a}} \tag{8}$$

$$\mathbf{y}_4 = k_\mathrm{a} = \mathbf{f}_4(\mathbf{x}) + \varepsilon_4 = k + \varepsilon_{k_\mathrm{a}} \tag{9}$$

where the subscript 'a' means the a priori value and $\varepsilon_{\cdots_\mathrm{a}}$ the a priori standard deviation, also acting as weight factors of the corresponding constraints. It has been proved in many studies that the $3\beta + 2\alpha$ measurements do not have enough sensitivity to accurately retrieve the CRI, especially to the imaginary part [20,21,23,43]. The introduction of the a priori constraints on CRI is in fact equivalent to prescribing a reasonable range for the retrieved CRI (centered at $n_\mathrm{a}$, $k_\mathrm{a}$ with spread of $\varepsilon_{n_\mathrm{a}}$ and $\varepsilon_{k_\mathrm{a}}$, respectively). This strategy is feasible in most cases because the aerosol type can be determined before the retrieval from lidar observations [44–47] and supplementary information (satellite data, atmospheric transport model, etc.) and type-resolved aerosol CRIs from in situ measurements or multi-angle passive remote sensing [48,49] are available.

For clarity, we rewrite Equations (5) and (7)–(9) into a uniform form:

$$\mathbf{y}_l = \mathbf{f}_l(\mathbf{x}) + \varepsilon_l, (l = 1, 2, 3, 4) \tag{10}$$

where $l = 1$ represents the equations describing the lidar measurements and $l = 2, 3, 4$ represent the equations about a priori constraints. If we assume $\varepsilon_l$ values are independent of each other and follow the Gaussian probability density function, the likelihood function [50] of the retrieval parameter vector **x** can be expressed as

$$L(\mathbf{x}) = \prod_l P(\mathbf{y}_l|\mathbf{x}) = \prod_l \frac{1}{(2\pi)^{n/2}|\mathbf{C}_l|^{1/2}} \exp\left\{-\frac{1}{2}[\mathbf{y}_l - \mathbf{f}_l(\mathbf{x})]^T \mathbf{C}_l^{-1}[\mathbf{y}_l - \mathbf{f}_l(\mathbf{x})]\right\} \tag{11}$$

where $P(\mathbf{y}_l|\mathbf{x})$ represents the conditional probability of $\mathbf{y}_l$, and $\mathbf{C}_l$ is the covariance matrix of $\varepsilon_l$. $|\cdot|$ represents the determinant operator. According to the MLE, the value of **x** maximizing the likelihood function is the maximum likelihood estimate of **x**, which is equivalent to minimizing the following cost function:

$$\chi^2(\mathbf{x}) = \sum_{l=1}^{4} [\mathbf{y}_l - \mathbf{f}_l(\mathbf{x})]^T \mathbf{C}_l^{-1}[\mathbf{y}_l - \mathbf{f}_l(\mathbf{x})] \tag{12}$$

In this way, the search of the retrieval parameter vector **x** is converted to an optimal problem. Since negative values of the retrieval parameters do not carry any physical meaning, we implement logarithmic transformation to avoid negative values in the retrieval parameters [36] and rewrite Equation (12) as below:

$$\chi^2(\mathbf{X}) = \sum_{l=1}^{4} [\mathbf{Y}_l - \mathbf{F}_l(\mathbf{X})]^T \mathbf{S}_l^{-1}[\mathbf{Y}_l - \mathbf{F}_l(\mathbf{X})] \tag{13}$$

where $\mathbf{X} = \ln \mathbf{x}$, $\mathbf{Y}_l = \ln \mathbf{y}_l$, and $\mathbf{F}(\mathbf{X}) = \ln\left[\mathbf{f}_l\left(e^{\mathbf{X}}\right)\right]$. The measurement variances after the transformation (i.e., the diagonal elements of $\mathbf{S}_l$) are related with their relative variances. For instance, in the term representing the lidar measurements ($l = 1$):

$$S_i = \ln\left[\tfrac{1}{2}\left(1 + \sqrt{1 + \tfrac{4C_i}{y_i^2}}\right)\right] \approx \tfrac{C_i}{y_i^2}, \quad (C_i \ll 1) \tag{14}$$

Note that by converting Equation (12) to Equation (13), we assume the measurements conform to the multivariate lognormal probability density function. For measurement noise of positively defined characteristics, this assumption is supported by both theoretical analysis and experimental results [51], and for a very small variance, lognormal distribution is almost equivalent to normal distribution.

*2.2. Optimization Procedure*

The minimization of Equation (13) is in fact a multi-term nonlinear least-square fitting weighted by the corresponding covariance matrices. It is implemented by the Levenberg–Marquardt iteration [52] as below:

$$\begin{aligned}
\mathbf{X}^{(u+1)} &= \mathbf{X}^{(u)} + \Delta\mathbf{X}^{(u)}, \\
\Delta\mathbf{X}^{(u)} &= \mathbf{G}_u^{-1}\mathbf{b}_u
\end{aligned} \tag{15}$$

where

$$\begin{aligned}
\mathbf{G}_u &= \sum_{l=1}^{4} \mathbf{J}_{l,\mathbf{X}^{(u)}}^{T}\mathbf{S}_l^{-1}\mathbf{J}_{l,\mathbf{X}^{(u)}} + \gamma^{(u)}\mathbf{D}, \\
\mathbf{b}_u &= \sum_{l=1}^{4} \mathbf{J}_{l,\mathbf{X}^{(u)}}^{T}\mathbf{S}_l^{-1}\left[\mathbf{Y}_l - \mathbf{F}_l\left(\mathbf{X}^{(u)}\right)\right]
\end{aligned} \tag{16}$$

and the superscript $(u)$ represents the $u$th iteration. $\mathbf{J}_{l,\mathbf{X}^{(u)}}$ is the Jacobian matrix of $\mathbf{F}_l$ at $\mathbf{X}^{(u)}$, $\mathbf{D}$ is a scaling matrix controlling the relative iteration steps, and $\gamma^{(u)}$ is the overall scalar factor controlling the speed of the convergence of the iteration process. The value of $\gamma^{(u)}$ should be adjusted in each iteration to ensure the reduction of the cost function and the non-singularity of $\mathbf{G}$. We adopt the following strategy to update $\gamma^{(u)}$ [36]:

$$\gamma^{(u)} = \frac{2\chi^2\left(\mathbf{X}^{(u)}\right)}{p - q} \tag{17}$$

where $p$ and $q$ are the number of total general measurements and the number of retrieval parameters and, in our case, $p = 13$ and $q = 10$.

A study of Veselovskii et al. [25] shows that both extinction kernels and backscattering kernels become highly interdependent and asymptotic to zero at large radii, which means the large particles of the size distribution contribute less to the total optical data. Thus, it is quite possible that the iteration can converge to an unrealistic but smooth monotonic VSD curve with large values at large radii, which simultaneously fits all the terms in Equation (13) quite well. We call such a VSD curve 'oversmoothed' and the cost function 'overfitted'. To avoid this issue, we set the stopping conditions to

1. $\chi^2\left(\mathbf{X}^{(u)}\right) < p - q$,
2. the number of iteration $u$ reaches the prescribed maximum value, and the iteration will stop if either of the above conditions is met. Condition 1 is based on the statistical principle. Since we have assumed each $\mathbf{Y}_l$ conforms to a Gaussian distribution, $\chi^2$ conforms to a chi-square distribution with a degree of freedom (DOF) of $p$–$q$. A 'good' fit is derived if the ratio of $\chi^2$ and DOF is just not greater than 1 [53].

Setting an initial guess near the solution could accelerate the speed of convergence compared with setting it arbitrarily. The type-resolved a priori value on CRI should not be far from the actual value if the type of the aerosol could be correctly identified before

retrieval. Thus, we set the initial guesses of $n, k$ to $n_a, k_a$, respectively. Correspondingly, the initial guess of VSD is set to a constant function derived by fitting $\alpha_{532}$.

### 2.3. The Selection of Individual Solutions

The optimization procedure gives a solution for a specific aerosol size range, i.e., $[r_{\min}, r_{\max}]$, which is called an inversion window hereafter. A solution corresponding to a specific inversion window is referred to as an individual solution. A proper inversion window covering the real aerosol size range is important for deriving a realistic solution. However, such a priori information is hardly available. Therefore, we decide to perform the inversion for a set of pre-defined inversion windows covering the radius range of 0.05–15 µm and then select the qualified individual solutions by some extra constraints. Due to the simultaneous retrieval of the VSD and CRI, there is only one individual solution for an inversion window rather than several hundred derived by linear methods [20,21,23], which simplifies the selection procedure.

Plenty of previous research reveals that, in most cases, the VSD of atmospheric aerosol ensembles conforms to multi-mode lognormal distributions [49,54–57]. Thus, we take this conclusion as an extra a priori constraint on VSD to select proper inversion windows (i.e., qualified individual solutions) because unproper inversion windows (either too wide or too narrow) can cause significant oscillations of the retrieved VSD curve in the wing zones, making it deviate from a 'lognormal' shape. However, such "deformed" curves can have very low fitting error due to the ill-posedness of the system. Thus, in addition to selecting qualified individual solutions by judging their fitting errors, we also select by judging whether the retrieved VSD has a lognormal-like shape. Specifically speaking, the selection procedure consists of the following steps:

1.  Select the individual solutions with fitting errors less than the prescribed measurement error (10% for all the measurement channels in this study);
2.  Among the selected individual solutions, select those whose elements of **v** meet either of the following inequalities:

$$
\begin{cases}
v_1 < v_2 \\
v_1 < 0.7 v_{\max} \\
v_8 < v_7 \\
v_8 < 0.7 v_{\max}
\end{cases}
\quad \text{or} \quad
\begin{cases}
v_1 > v_2 \\
v_1 < 0.05 v_{\max} \\
v_8 > v_7 \\
v_8 < 0.05 v_{\max}
\end{cases}
\tag{18}
$$

where $v_{\max}$ means the maximum retrieved element in **v,** and the multiple factors are empirically chosen.

3.  Among the selected individual solutions, select those whose standard deviations of the VSD are greater than 0.35. This criterion is based on the study of Tanré et al. [58]. The standard deviation of a distribution $v\,(\ln r)$ is calculated by:

$$
\sigma_{\mathrm{v}} = \sqrt{\frac{\int_{r_{\min}}^{r_{\max}} (\ln r - \ln \mu)^2 v(\ln r) d \ln r}{\int_{r_{\min}}^{r_{\max}} v(\ln r) d \ln r}}
\tag{19}
$$

where

$$
\mu = \exp\left[\frac{\int_{r_{\min}}^{r_{\max}} \ln r \cdot v(\ln r) d \ln r}{\int_{r_{\min}}^{r_{\max}} v(\ln r) d \ln r}\right]
\tag{20}
$$

After determining the "qualified" individual solutions, we average them (the average of both retrieved VSDs and retrieved CRIs) to build the final averaged solution, which is regarded as the retrieval result of the case.

In addition, to describe the bulk properties of a particle ensemble, total volume concentration ($V_t$) and effective radius ($R_{\mathrm{eff}}$) can be calculated from the retrieved VSD:

$$
V_{\mathrm{t}} = \int_{r_{\min}}^{r_{\max}} v(\ln r) d \ln r
\tag{21}
$$

$$R_{\text{eff}} = \frac{\int_{r_{\min}}^{r_{\max}} v(\ln r) d \ln r}{\int_{r_{\min}}^{r_{\max}} \frac{1}{r} v(\ln r) d \ln r} \tag{22}$$

*2.4. Propagation of Measurement Error*

In this part, we evaluate the influence of lidar measurement error on individual solutions. According to Equation (15), if the iteration stops at $u$, we have

$$\hat{\mathbf{X}} = \mathbf{X}^{(u)} = \mathbf{X}^{(u-1)} + \Delta \mathbf{X}^{(u-1)} \tag{23}$$

where $\hat{\mathbf{X}}$ means the retrieved value of $\mathbf{X}$. If the variation of $\hat{\mathbf{X}}$ due to a lidar measurement error $d\mathbf{Y}_1$ can be approximated to be linear, we derive

$$\frac{d\hat{\mathbf{X}}}{d\mathbf{Y}_1} = \frac{\partial \mathbf{X}^{(u)}}{\partial \mathbf{Y}_1} = \frac{\partial \mathbf{X}^{(u-1)}}{\partial \mathbf{Y}_1} + \frac{\partial \Delta \mathbf{X}^{(u-1)}}{\partial \mathbf{Y}_1} \tag{24}$$

According to the rules of nested matrix calculus, we have

$$\begin{aligned}
\frac{\partial \Delta \mathbf{X}^{(u-1)}}{\partial \mathbf{Y}_1} &= \left( \mathbf{G}_{u-1}^{-1} \mathbf{b}_{u-1} \right)^T \otimes \left( -\mathbf{G}_{u-1}^{-1} \right) vec(\mathbf{D}) \left( \frac{\partial \gamma^{(u-1)}}{\partial \mathbf{Y}_1} \right)^T \\
&+ \mathbf{G}_{u-1}^{-1} \left[ \mathbf{J}_{1,\mathbf{X}^{(u-1)}}^T \mathbf{S}_1^{-1} - \left( \sum_{l=1}^{4} \mathbf{J}_{l,\mathbf{X}^{(u-1)}}^T \mathbf{S}_l^{-1} \mathbf{J}_{l,\mathbf{X}^{(u-1)}} \right) \frac{\partial \mathbf{X}^{(u-1)}}{\partial \mathbf{Y}_1} \right]
\end{aligned} \tag{25}$$

where

$$\frac{\partial \gamma^{(u-1)}}{\partial \mathbf{Y}_1} = \frac{4}{3} \left\{ \mathbf{S}_1^{-1} \left[ \mathbf{Y}_1 - \mathbf{F}_1 \left( \mathbf{X}^{(u-1)} \right) \right] - \sum_{l=1}^{4} \left( \mathbf{J}_{l,\mathbf{X}^{(u-1)}} \frac{\partial \mathbf{X}^{(u-1)}}{\partial \mathbf{Y}_1} \right)^T \mathbf{S}_l^{-1} \left[ \mathbf{Y}_l - \mathbf{F}_l \left( \mathbf{X}^{(u-1)} \right) \right] \right\} \tag{26}$$

The operator $\otimes$ represents the Kronecker product of two matrices and $vec(\cdot)$ means the vectorization of a matrix [59]. With Equations (25) and (26), we can calculate $d\hat{\mathbf{X}}/d\mathbf{Y}_1$ iteratively and note that $d\mathbf{X}^{(0)}/d\mathbf{Y}_1 = 0$. Correspondingly, the covariance matrix of $\hat{\mathbf{X}}$, denoted as $\hat{\mathbf{S}}$, can be calculated from

$$\hat{\mathbf{S}} = \left( \frac{d\hat{\mathbf{X}}}{d\mathbf{Y}_1} \right) \mathbf{S}_1 \left( \frac{d\hat{\mathbf{X}}}{d\mathbf{Y}_1} \right)^T \tag{27}$$

and since $\hat{\mathbf{x}} = \exp \hat{\mathbf{X}}$, the variation and covariance matrix of $\hat{\mathbf{x}}$, denoted as $d\hat{\mathbf{x}}$ and $\hat{\mathbf{C}}$, respectively, are

$$d\hat{\mathbf{x}} = \exp \hat{\mathbf{X}} d\hat{\mathbf{X}} \tag{28}$$

$$\{\hat{\mathbf{C}}\}_{ij} = E(\hat{x}_i) E(\hat{x}_j) \left[ \exp\{\hat{\mathbf{S}}\}_{ij} - 1 \right] \tag{29}$$

where $E(\hat{x}_i) = \exp\left( \hat{X}_i + \{\hat{\mathbf{S}}\}_{ii}/2 \right)$ is the expectation of the $i$th element of $\hat{\mathbf{x}}$ and $E(\hat{x}_j)$ the expectation of the $j$th element. Likewise, the variety and covariance matrices of $V_t$ and $R_{\text{eff}}$ can be calculated from

$$dI = \frac{dI}{d\hat{\mathbf{x}}} d\hat{\mathbf{x}}, \quad (I = V_t, R_{\text{eff}}) \tag{30}$$

$$\mathbf{C}_I = \left( \frac{dI}{d\hat{\mathbf{x}}} \right) \mathbf{C}_{\hat{\mathbf{x}}} \left( \frac{dI}{d\hat{\mathbf{x}}} \right)^T, \quad (I = V_t, R_{\text{eff}}) \tag{31}$$

We are interested in deriving these above relations because they facilitate both the calculation of retrieval sensitivity in sensitivity study and the calculation of retrieval uncertainty in real application. However, their accuracies depend on the linearity of the system when lidar measurements vary in a range of measurement errors. In the next section, we will examine the feasibility of these relations by numerical simulations.

## 3. Sensitivity Study

The first part of this section is focused on assessing the performance of the BOREAL algorithm by inverting synthetic optical data generated by different aerosol models. We derive retrieval results for these aerosol models with and without considering measurement error, respectively, and compare them with their original values. In the second part of this section, we evaluate the feasibility of the error propagation model proposed in Section 2.4.

### 3.1. Data Preparation and Initialization

As indicated in Section 2.3, we use the lognormal distribution to model the VSD of aerosols composed of spherical particles:

$$v(\ln r) = \frac{dV_{\mathrm{t}}}{d \ln r} = \sum_{i=\mathrm{f,c}} \frac{dV_i}{\sqrt{2\pi}\sigma_{\mathrm{v},i}} \exp\left[ -\frac{(\ln r - \ln r_{\mathrm{v},i})^2}{2\sigma_{\mathrm{v},i}^2} \right] \tag{32}$$

where the subscript $i$ indicates the fine mode ($i$ = f) or coarse mode ($i$ = c). In each mode, $V_i$ represents the volume concentration, $\sigma_{\mathrm{v},i}$ the geometric standard deviation, and $r_{\mathrm{v},i}$ the mode radius. $V_{\mathrm{t}}$ is the total volume concentration, the same parameter defined by Equation (21).

According to previous characterization of aerosol types [30,49,60–63], we assumed 4 types of VSDs and 25 spectrally independent CRIs, as shown in Table 1. Synthetic optical data ($3\beta + 2\alpha$), which are to be inverted with BOREAL, were calculated from these aerosol models with the Mie theory using the databank of Dubovik et al. [37].

**Table 1.** Aerosol models used for generating synthetic ($3\beta + 2\alpha$) data. The definitions of the parameters describing the lognormal VSD can be found in Equation (32). Four VSD types (MF for mono-fine mode, MC for mono-coarse mode, BF for bimodal with fine-mode-dominant, and BC for bimodal with coarse-mode-dominant) and 25 combinations of complex refractive index (CRI = $n - ik$) are prescribed.

| SD Type | $V_{\mathrm{f}}$ | $r_{\mathrm{v,f}}$ | $\sigma_{\mathrm{v,f}}$ | $V_{\mathrm{c}}$ | $r_{\mathrm{v,c}}$ | $\sigma_{\mathrm{v,c}}$ | $V_{\mathrm{t}}$ | $R_{\mathrm{eff}}$ |
|---------|------|-------|--------|------|-------|--------|------|--------|
| MF | 1 | 0.2 | 0.4 | 0 | 0 | 0 | 1 | 0.18 |
| MC | 0 | 0 | 0 | 1 | 1.2 | 0.6 | 1 | 0.99 |
| BF | 2/3 | 0.2 | 0.4 | 1/3 | 2 | 0.6 | 1 | 0.26 |
| BC | 1/6 | 0.2 | 0.4 | 5/6 | 2 | 0.6 | 1 | 0.70 |
| $n_{\mathrm{ture}}$ | | | | 1.4, 1.45, 1.5, 1.55, 1.6 | | | | |
| $k_{\mathrm{true}}$ | | | | 0.001, 0.005, 0.01, 0.015, 0.02 | | | | |

We use $(n_{\mathrm{a}}, \varepsilon_{n_{\mathrm{a}}}) = (1.5, 0.1)$ as the a priori constraint on the real part of the CRI for all the cases; $(k_{\mathrm{a}}, \varepsilon_{k_{\mathrm{a}}}) = (0.005, 0.005)$ for non-absorbing cases, where $k_{\mathrm{true}} \leq 0.01$; and $(k_{\mathrm{a}}, \varepsilon_{k_{\mathrm{a}}}) = (0.015, 0.01)$ for absorbing cases, where $k_{\mathrm{true}} > 0.01$. We will also use this configuration for inverting real lidar measurements before an applicable aerosol typing method is developed and a correlated type-resolved database of the a priori constraints is established.

### 3.2. Evaluation of Retrieval Accuracy

Figure 1 shows the comparisons between the retrieved and true VSDs when $n_{\mathrm{ture}} = 1.6$ and $k_{\mathrm{ture}} = 0.01$. The left column (Figure 1a1–d1) represents the results when the synthetic optical data were free of error (error-free), while the right column (Figure 1a2–d2) shows the statistics of the results when measurement error is considered (error-contaminated), which is accomplished by adding the error vector to the optical data and inverting the error-contaminated optical data 100 times. The elements of the error vector are independent of each other and conform to the Gaussian distribution: $\sim N(0, 0.1)$. From Figure 1, it is seen that there are larger dispersions in the coarse mode than in the fine mode for both error-free and error-contaminated optical data. This can be explained by the fact

that the backscattering kernels decrease rapidly if the particle radius exceeds 2–3 μm [25], which undermines the contribution of the coarse mode to total backscattering when both modes exist.

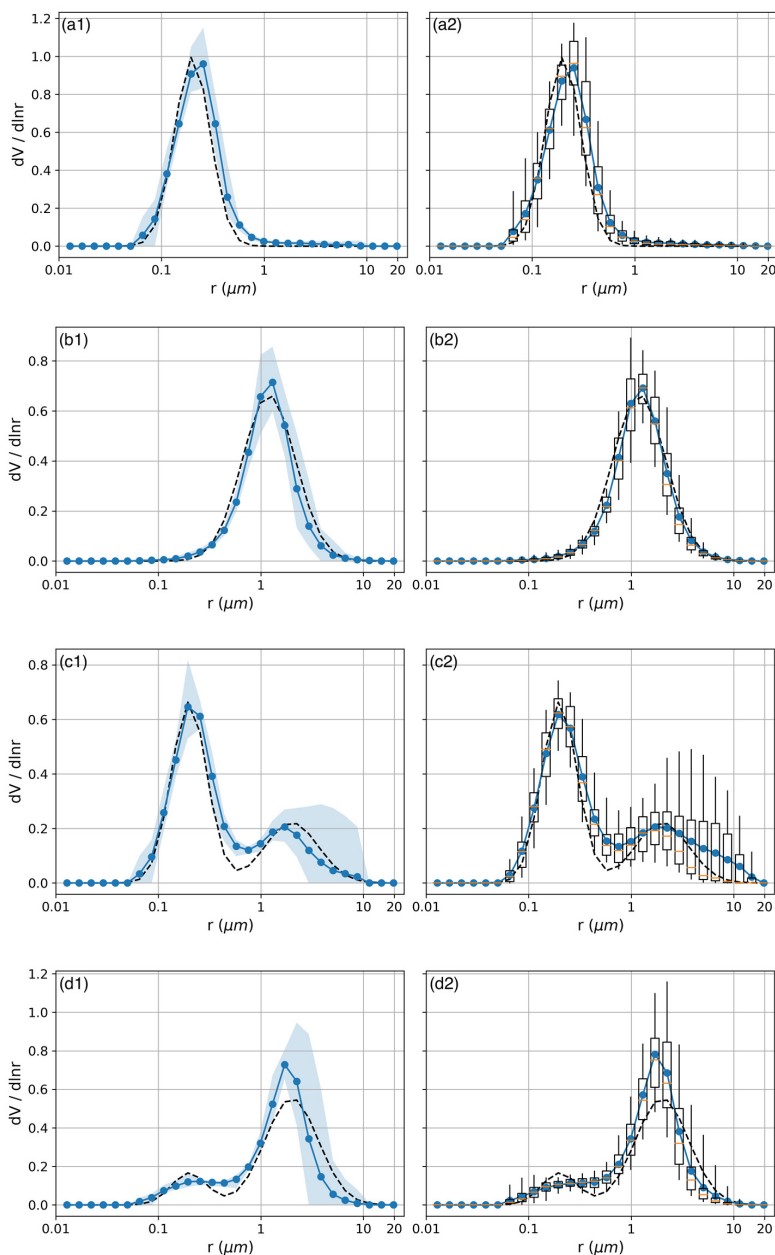

**Figure 1.** Comparisons of the original volume size distributions (VSDs) of the aerosol models and the retrieved VSDs. Four different VSDs in Table 1. (a. MF, b. MC, c. BF, d. BC) with complex refractive index (CRI) equal to 1.6 -i0.01 were considered. The left column (**a1**–**d1**) corresponds to the error-free optical data, where the true VSDs (dashed lines), upper and lower limits of the selected individual solutions (shaded areas), and the averaged solutions (circle solid lines) are shown. The right column (**a2**–**d2**) represents the statistics of the results when measurement error is considered, which is accomplished by adding the Gaussian error to the optical data and inverting the error-contaminated optical data 100 times. The box-and-whiskers plots show the distribution of the retrieval results, where the endpoints and horizontal lines from bottom to top correspond to the values below which 5%, 25%, 50%, 75%, and 95% of the results lie (namely, the percentile of the statistics). The blue solid lines connect the mean values of each bin.

Table 2 shows the retrieval differences, defined as the difference between the retrieved value and true value, in CRIs, $V_t$, and $R_{eff}$ corresponding to the scenarios presented in Figure 1. For all these scenarios, both the real part and imaginary part are underestimated by approximately 0.05 and 50%, respectively. The retrieved imaginary parts are quite close to the a priori value ($k_a = 0.005$ for $k_{true} \leq 0.01$), and the retrieved real parts lie in the range $[n_a, n_{true}]$. If the a priori constraint $(k_a, \varepsilon_{k_a}) = (0.015, 0.01)$ is used, the imaginary parts for all the cases will be overestimated, with retrieved values also near $k_a$ (not shown here). These facts indicate the influence of a priori constraints on the retrieval of CRI, especially on the imaginary part. To the contrary, the change of a priori constraints hardly affects the retrieval of VSDs for these scenarios. From Table 2, the retrievals of the VSDs for these scenarios have similar accuracies with $\delta V_t$ ranging in $[-8\%, -13\%]$ and $\delta R_{eff}$ ranging in $[-4\%, -11\%]$ if the optical data are error-free. On the other hand, measurement error affects these retrievals in two aspects. Firstly, it causes bias in some parameters, for example, the $V_t$, $R_{eff}$ of Type BF, and Type BC. Such bias results from the overestimate of the VSD of the coarse mode, which can be inferred from Figure 1. Secondly, it disperses the retrieved parameters to a different extent, acting as statistical standard deviations shown in Table 2: the magnitudes of dispersions in $k$, $V_t$, $R_{eff}$ are comparable with the measurement error, while those in $n$ are much less than the measurement error.

**Table 2.** Retrieval differences [1], defined as the difference between the retrieved value and true value, in $n$, $k$, $V_t$, and $R_{eff}$, for the scenarios presented in Figure 1. For the error-contaminated optical data, mean differences and standard deviations (in parentheses) of the statistics are shown.

| | Error-Free Optical Data | | | | Error-Contaminated Optical Data | | | |
|---|---|---|---|---|---|---|---|---|
| | $\delta n$ | $\delta k$ | $\delta V_t$ | $\delta R_{eff}$ | $\delta n$ | $\delta k$ | $\delta V_t$ | $\delta R_{eff}$ |
| MF | $-0.05$ | $-53\%$ | 13% | 11% | $-0.05$ (2%) | $-52\%$ (10%) | 16 (11%) | 11% (15%) |
| MC | $-0.03$ | $-49\%$ | $-8\%$ | $-4\%$ | $-0.03$ (1%) | $-51\%$ (8%) | $-9\%$ (12%) | $-6\%$ (12%) |
| BF | $-0.05$ | $-49\%$ | 6% | 4% | $-0.05$ (2%) | $-47$ (9%) | 24% (19%) | 15% (23%) |
| BC | $-0.06$ | $-44\%$ | 4% | $-4\%$ | $-0.06$ (1%) | $-46\%$ (9%) | 10% (22%) | 0% (26%) |

[1] The retrieval differences in $n$ are in absolute values, while those in $k$, $V_t$, and $R_{eff}$ are in relative values.

Figure 2 shows the statistics of the absolute retrieval differences (the absolute value of retrieval difference, which is always positive) of CRIs, $V_t$, and $R_{eff}$ for all the scenarios in Table 1. In general, compared with other aerosol types, retrieval differences for MF aerosols have the lowest medium values and smallest dispersions, representing the best retrieval accuracies among the four VSD types. On the other hand, BC aerosols have the largest dispersions in $\delta V_t$ and $\delta R_{eff}$, which likely results from the errors in coarse-mode retrieval. For different retrieval parameters, measurement error enlarges the retrieval dispersion to various extents, influencing $V_t$ and $R_{eff}$ more than $n$ and $k$. In particular, the dispersions of $\delta k$ are nearly the same with and without measurement error. Figure 3 shows in detail the distribution of $\delta k$, from which we see $\delta k > 300\%$ when $k_{true} = 0.001$, regardless of the VSD type and $n_{true}$. This is because in these scenarios, the retrieved values of $k$ are all close to $k_a$. Such retrieval difficulty is also faced in linear regularization methods [62]. Retrieval accuracy of $k$ improves with the increase of $k_{true}$ and with $n_{true}$ getting close to $n_a$. For example, $\delta k$ smaller than 10% can be derived when $k_{true} = 0.02$ and $n_{true} = n_a$ for all types of size distributions.

Table 3 summarizes the third quartiles of the retrieval differences corresponding to Figure 2, which we adopt as an overall estimate of retrieval accuracy with respect to the VSD type. The sensitivity study shows that using the configuration in Section 3.1, the values of VSD, $V_t$, $R_{eff}$, and CRI for typical aerosols could be retrieved with acceptable accuracies by BOREAL in the case of relative measurement uncertainty in each channel less than 10%. Note that the last quartile of $\delta k$ corresponds to the scenarios where $k_{true} = 0.001$, which are all above 300%, according to Figure 3. Accordingly, once again, we emphasize the importance of the a priori information on CRI, especially on the imaginary part, to

effectively constrain the final solution. Retrieval accuracy for monomodal aerosols is comparable to the result of Müller et al. [21], where a linear inversion algorithm was used to retrieve $V_t$, $R_{eff}$, and CRI.

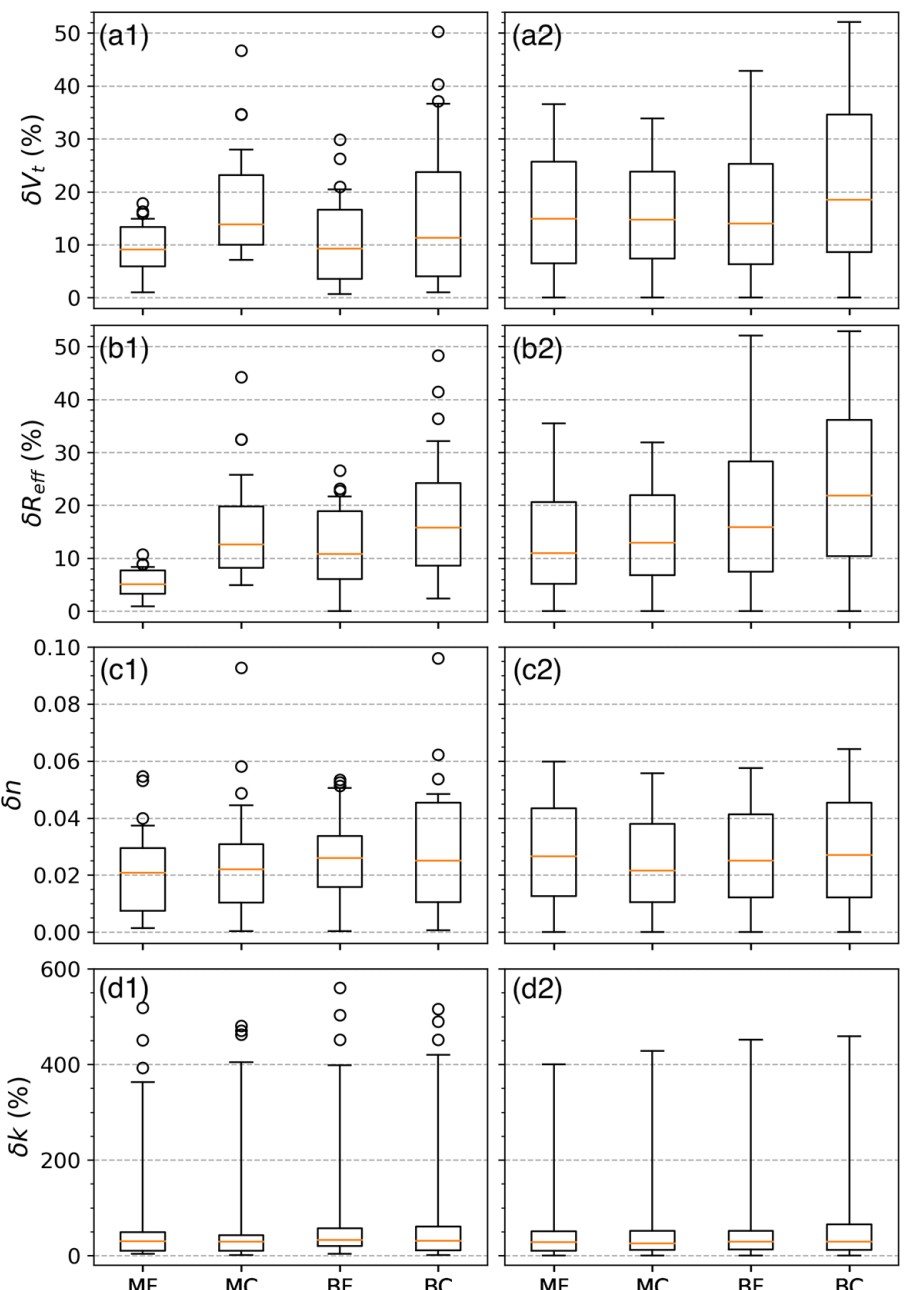

**Figure 2.** Box-and-whisker plots of retrieval differences, defined as the difference between the retrieved value and true value, in $V_t$ (%), $R_{eff}$ (%), $n$, and $k$ (%) with respect to the VSD types for all the scenarios in Table 1. The left column (**a1–d1**) corresponds to the error-free optical data and the right column (**a2–d2**) to the error-contaminated optical data (i.e., each error-free scenario is perturbed by Gaussian error 100 times, thus, 10,000 scenarios in total). The hinges and horizontal lines from the bottom to top of the box-and-whiskers plots successively represent the 0, 25, 50, 75, and 90 percentiles of the dataset. Data beyond the top hinge are designated outliers and shown as hollow circles. Considering the size of the dataset, the outliers corresponding to the error-contaminated optical data are not shown.

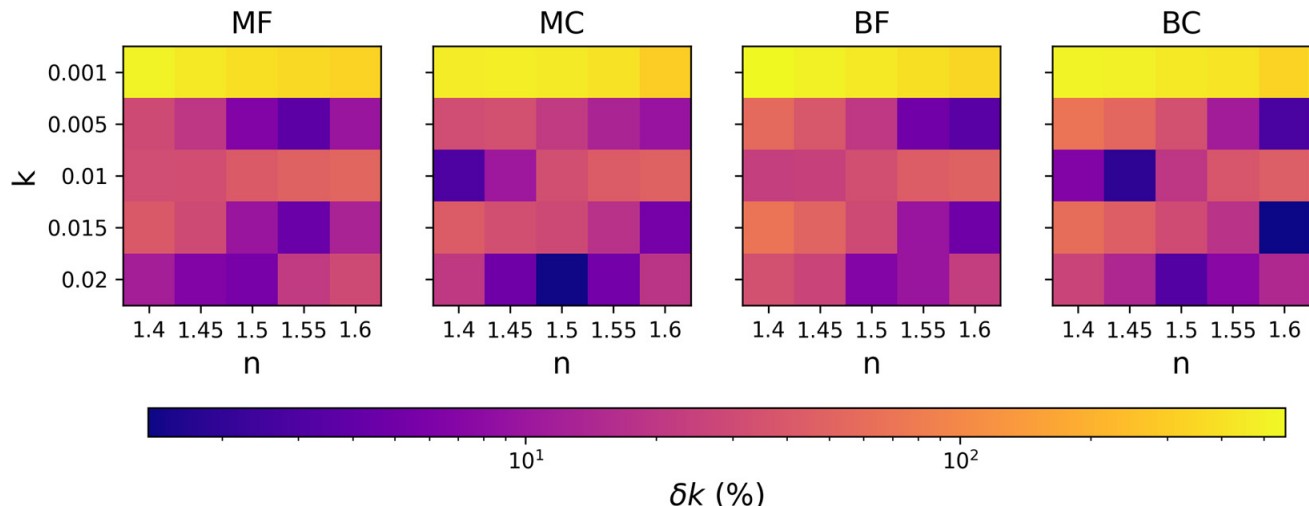

**Figure 3.** Distribution of $\delta k$ for the retrieval scenarios in Table 1.

**Table 3.** Third quartiles of $\delta V_{\mathrm{t}}$, $\delta R_{\mathrm{eff}}$, $\delta n$, and $\delta k$ corresponding to Figure 2.

|  | Error-Free Optical Data | | | | Error-Contaminated Optical Data | | | |
|---|---|---|---|---|---|---|---|---|
|  | $\delta V_{\mathrm{t}}$ | $\delta R_{\mathrm{eff}}$ | $\delta n$ | $\delta k$ | $\delta V_{\mathrm{t}}$ | $\delta R_{\mathrm{eff}}$ | $\delta n$ | $\delta k$ |
| MF | 13% | 8% | 0.030 | 49% | 26% | 21% | 0.045 | 51% |
| MC | 24% | 19% | 0.031 | 43% | 24% | 22% | 0.038 | 52% |
| BF | 18% | 16% | 0.034 | 55% | 25% | 28% | 0.040 | 52% |
| BC | 23% | 19% | 0.042 | 55% | 35% | 36% | 0.045 | 65% |

### 3.3. Evaluation of the Error Propagation Model

In the second part of this section, we evaluate the feasibility of the error propagation model proposed in Section 2.4. Note that in this subsection, all the retrieval parameters are with respect to the individual solution.

Firstly, we evaluate when $\hat{x}$, the function of lidar measurement $\mathbf{y}_1$, could be approximated to be linear as $\mathbf{y}_1$ varying in $\mathbf{y}_1 \pm \boldsymbol{\varepsilon}_1$. To this end, we define the relative approximation error (RAE) of a single retrieval parameter as

$$\rho = \left| \frac{\hat{x}_{\mathrm{pa}} - \hat{x}_{\mathrm{p}}}{x - \hat{x}_{\mathrm{p}}} \right| \tag{33}$$

where $x$ is the true value of the retrieval parameter, $\hat{x}_{\mathrm{p}}$ is the retrieved value when a known perturbation is added to $\mathbf{y}_1$, and

$$\hat{x}_{\mathrm{pa}} = \hat{x} + d\hat{x} \tag{34}$$

where $\hat{x}$ is the retrieved value when no perturbation is added to $\mathbf{y}_1$, and $d\hat{x}$ is calculated through the equations in Section 2.4. A low $\rho$ indicates the linearization error is minor compared with the retrieval error caused by lidar measurement error and algorithmic error. In general, RAE should increase with the increase of measurement error because it could substantially change the path of the minimization procedure, for example, changing the iteration number from $u$ to $u'$, which enlarges the difference between $\hat{x}_{\mathrm{p}}$ and $\hat{x}_{\mathrm{pa}}$ since $d\hat{x}$ is evaluated for the iteration number $u$ rather than $u'$.

For the scenarios in Table 1, we assigned suitable inversion windows corresponding to their VSDs. Then, we perform retrieval and calculate the RAEs of $V_{\mathrm{t}}$, $R_{\mathrm{eff}}$, $n$, and $k$ when the error-free optical data are perturbed by 1%, 5%, and 10%, respectively. Optical data at different wavelengths are perturbed by the same magnitude but with different signs to imitate random effects, as shown in Table 1 of [21]. Figure 4 shows the statistical results for the MF aerosol, which are classified by whether the iteration number changes. As discussed

above, RAEs for the scenarios where the iteration number changes are 3–5 times higher than those where the iteration number does not change. At the same time, the number of scenarios where the iteration number changes increases with the increase of the magnitude of perturbation. For a measurement uncertainty of 10% in each channel, (1) more than 80% of the scenarios have their iteration numbers changed with quite large RAEs and (2) among the scenarios with unchanged iteration numbers, more than 50% have the RAEs greater than 0.3, 0.4, 0.1, and 0.1 in $V_t$, $R_{eff}$, $n$, and $k$, respectively.

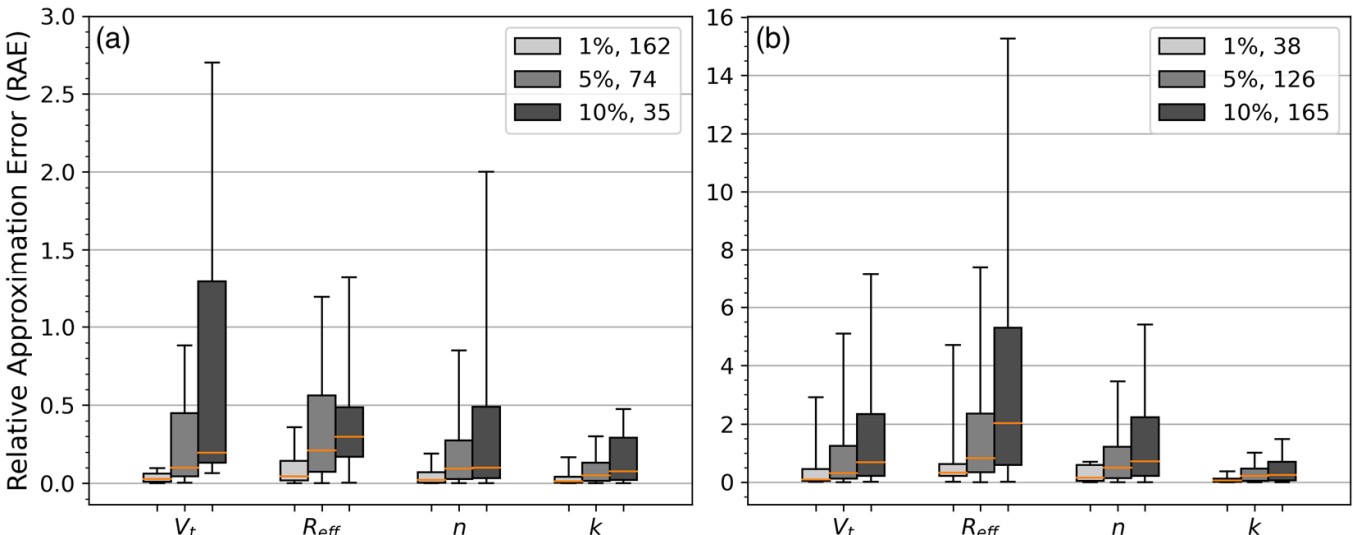

**Figure 4.** Relative approximation error (RAE) of $V_t$, $R_{eff}$, $n$, and $k$ for MF aerosols in Table 1. (**a**) The results of which the iteration number does not change after the introduction of perturbation; (**b**) the results of which the iteration number changes after the introduction of perturbation. The magnitudes of perturbations (1%, 5%, and 10%) are labeled in the legend, followed by the counts of the cases. The hinges and horizontal lines from the bottom to top of the box-and-whiskers plots represent 0, 25, 50, 75, and 90 percentiles of the dataset.

Then, under inversion windows the same as those mentioned above, we evaluate the retrieval standard deviation (RStd) calculated with the error propagation model for a measurement uncertainty of 10%. Figure 5 shows a case-by-case comparison of the RStds of $V_t$, $R_{eff}$, $n$, and $k$ calculated with the error propagation model (*y*-axes) and derived from the statistics of the 100 inversions of error-contaminated optical data (same as the method described in Section 3.2) (*x*-axes). From Figure 5, it is seen that the correlation of the RStd depends on the retrieval parameter and VSD type and, generally speaking, the difference between the calculation and experimental result is too large to allow the error propagation model to be applicable for estimating the retrieval uncertainty of the individual solution under 10% measurement uncertainty.

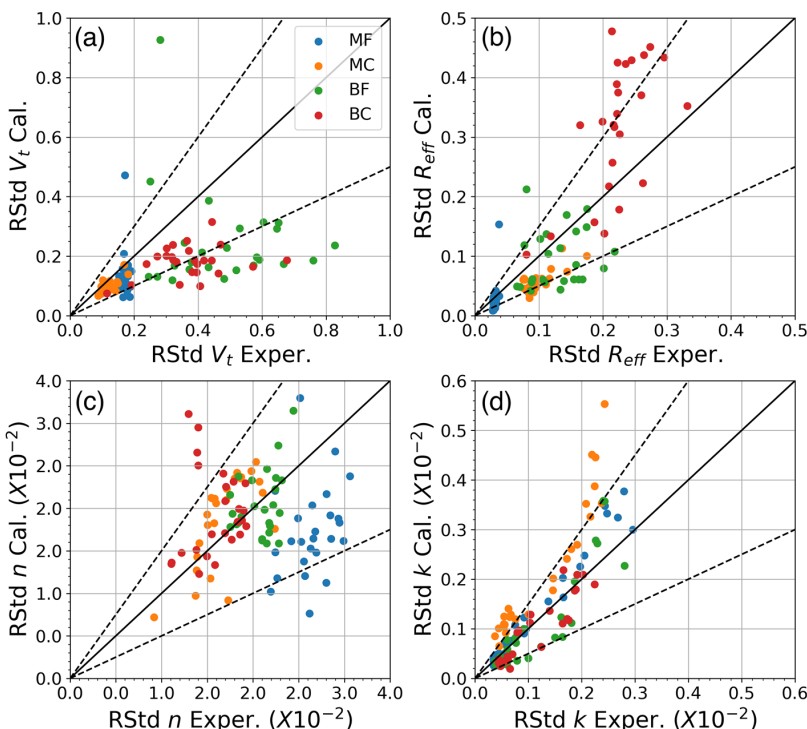

**Figure 5.** Case-by-case comparison of the retrieval standard deviation (RStd) of (**a**) $V_t$, (**b**) $R_{eff}$, (**c**) $n$, and (**d**) $k$ calculated with the propagation model for a measurement uncertainty of 10% ($y$-axes) and derived from the statistics of the 100 inversions of error-contaminated optical data (same as the method described in Section 3.2) ($x$-axes). For each VSD type, individual solutions are derived for suitable inversion windows. In each panel, the black solid line represents the 1–1 line, and between the two dashed lines is the area where relative error is less than 50%.

## 4. Application to Real Lidar Measurements

To test the algorithmic performance on real aerosol events, we applied BOREAL algorithm to three representative aerosol events detected by LILAS (LIlle Lidar AtmosphereS). LILAS is a high-performance Mie–Raman–Fluorescent lidar system developed at Laboratoire d'Optique Atmosphérique as of 2013. It is capable of measuring $3\beta + 2\alpha + 3\delta + 1\beta_F$ simultaneously, where "$3\delta$" is referred to as the particle depolarization ratio at 355 nm, 532 nm, and 1064 nm, while "$1\beta_F$" means the fluorescent backscattering coefficient centered at 466 nm. Detailed descriptions regarding the instrument and measurement uncertainties can be found in Hu et al. [30] and Veselovskii et al. [31]. The computer used for the retrievals is equipped with a 2.3 GHz Intel 8-Core i9 processor. Processing time of the CPU in each case was counted as an indicator of the algorithmic efficiency.

### 4.1. Case 1: 10 April 2015, Dakar

This observation was recorded in Dakar during SHADOW-2 (study of Saharan Dust Over West Africa) campaign in 2015. According to the analysis of Veselovskii et al. [64], on 10 April, dry dust transported from the Sahara Desert was dominant in the atmosphere. Here, we retrieved the aerosol properties in the period of 00:00–02:00 UTC using BOREAL and compared them with the results presented in Veselovskii et al. [64], where the regularization method [38] was used to retrieve the aerosol microphysical properties. Since the spheroids' volume fraction (SVF) on that day was higher than 98%, according to AERONET retrieval, we assumed the particles were totally spheroidal, which was also adopted in Veselovskii et al. [64].

Figure 6 shows the comparison of aerosol optical parameters from lidar measurements in the period of 00:00–02:00 UTC, 10 April 2015, and recalculated from the retrieval of BOREAL. The layer 1500–4400 m, where mineral dust was mainly concentrated, was

selected and resampled for the retrieval. The total processing time was ~1 min. The overall difference between the lidar measurements and recalculated measurements was less than 10% for $\alpha$ and 5% for $\beta$. Figure 7 shows the comparison of the profiles of $V_t$, $R_{eff}$, and CRI retrieved by BOREAL and presented in Veselovskii et al. [64]. The $V_t$ and $R_{eff}$ derived from BOREAL were generally smaller but within the ranges of retrieval uncertainty provided by Veselovskii et al. [64]. The profiles of the real parts of the CRI, in Figure 6b are in good agreement. The increase of the extinction Angstrom exponent (EAE) and decrease of $\alpha$ indicate that particles became smaller and less concentrated upon 3300 m, which is reflected in $V_t$ and $R_{eff}$ in Figure 7.

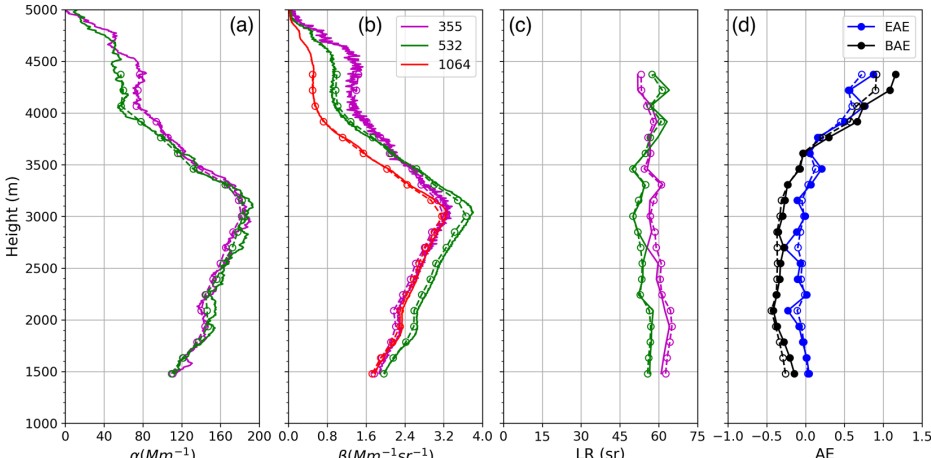

**Figure 6.** LILAS measurements (solid lines) and the measurements recalculated from the retrievals (dashed lines) on 10 April 2015, in the period of 00:00–02:00 UTC, at Dakar. (**a**) Extinction coefficients ($\alpha$); (**b**) backscattering coefficients ($\beta$); (**c**) Lidar ratios (LRs), and (**d**) Angstrom exponents of 355 nm over 532 nm (AE$_{355-532}$), including extinction Angstrom exponent (EAE$_{355-532}$) and backscattering Angstrom exponent (BAE$_{355-532}$). The layer 1500–4400 m was selected and resampled for the retrieval. Measurements at different wavelengths are represented by the corresponding colors.

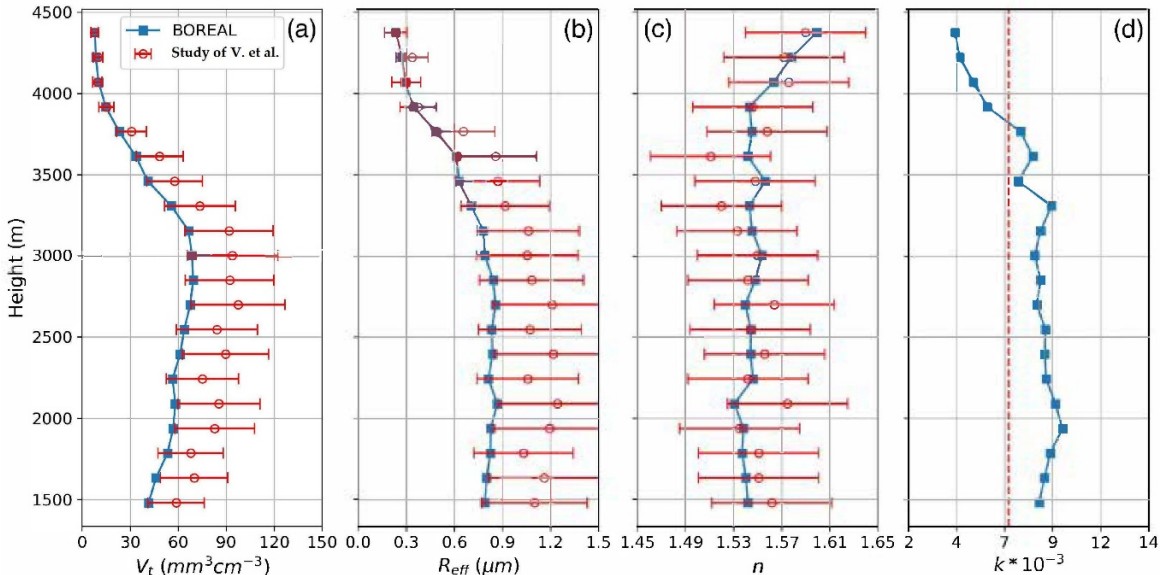

**Figure 7.** Comparison of retrieval results derived by BOREAL from Figure 6 (blue solid lines) and presented in Veselovskii et al. [64] (red hollow circles). (**a**) $V_t$; (**b**) $R_{eff}$; (**c**) $n$, and (**d**) $k$. The study in Veselovskii et al. [64] did not provide the profile of $k$ but an approximated value of 0.007 for the whole dust layer (red dashed line). Because the particles are all assumed to be spheroids, results in Table 3 cannot be used here as estimates of retrieval accuracies.

To further investigate possible reasons for the underestimation of $V_t$ and $R_{eff}$ compared with the results in Veselovskii et al. [64], Figure 8a shows the retrieved VSDs at two heights where aerosols were concentrated. For each level, our result shows a strict mono-coarse mode with $r_v \approx 1\,\mu m$, while an extra mode with $r_v \approx 3\,\mu m$ is shown in Veselovskii et al. [64]. We attribute such differences to algorithmic principles. Due to the optimal searching strategy, BOREAL derives only one individual solution for a specific inversion window. Nevertheless, the linear regularization method [64] retrieves VSD for every combination of CRI and inversion window pre-defined in the searching domain. In addition, differences between inversion windows and selection criteria between the two algorithms could also explain the different final averaged solutions. Due to the underdetermination in lidar inverse problems, it is hard to judge which retrieval is closer to the true state without the comparison with appropriate in situ measurements, which is indeed needed for further validation. However, by checking the fitting errors shown in Figure 6, we argue that the BOREAL-derived retrieval is reasonable enough for reproducing $3\beta + 2\alpha$ lidar measurements. The right panel of Figure 8 shows a comparison of the VSDs retrieved from the vertical-integrated LILAS measurements and from AERONET. The two retrievals both have a single coarse mode with quite similar $V_t$. However, LILAS/BOREAL retrieval has smaller $r_v$ and $R_{eff}$ possibly due to: (1) the influence of retrieved CRI: the LILAS/BOREAL retrieval gives a spectral independent CRI of $n = 1.55$ and $k = 0.009$, while the AERONET retrieval gives a spectral independent $n$ of 1.6 and a spectral dependent $k$ (decreasing from slightly above 0.004 to below 0.001 with the increase of wavelength); (2) contributions of aerosols in the boundary layer are not taken into account in LILAS retrieval; and (3) temporal difference of 7 h between the two retrievals.

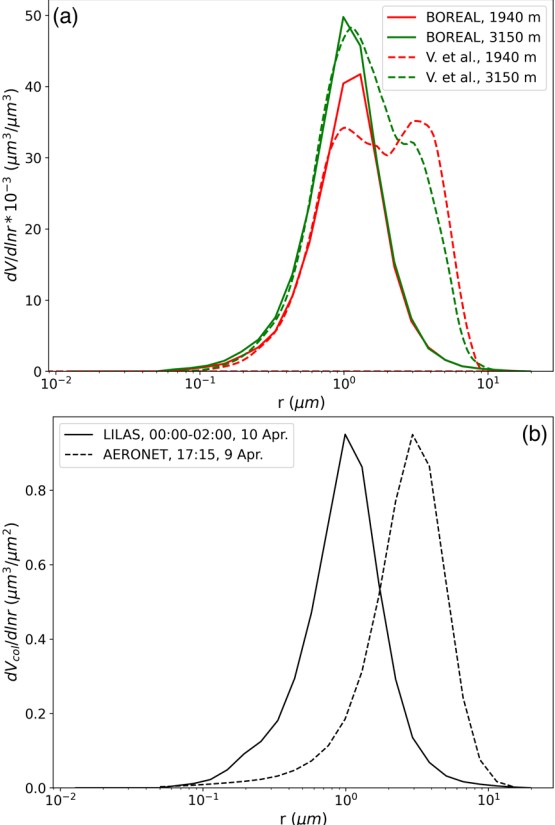

**Figure 8.** Comparison of VSD retrieval. (**a**) Comparison between the VSDs retrieved by BOREAL (solid lines) and presented in Veselovskii et al. [64] (dashed lines) at 2 concentrated levels, the "\*" in the label of the ordinate means the multiplication symbol; (**b**) VSDs retrieved from the vertical-integrated LILAS measurements (1500–4500 m, solid line) and from AERONET measurement at 17:15 UTC, 9 April (dashed line).

*4.2. Case 2: 11–12 September 2020, Lille*

During this period, aged biomass burning aerosols (BBA) originating from California wildfires were observed by LILAS in operation at ATOLL [65]. Here, we averaged the measurements in the period of between 22:30–03:00 UTC, 11–12 September 2020, and retrieved the layer 5000–9000 m. Note that the vertical resolution in this case was reduced to 500 m due to the low SNR in the upper troposphere. Spherical and absorbing particle assumptions were used in the retrieval.

Figure 9 shows the LILAS measurements and the recalculated measurements in that period. The total processing time was ~1 min. The overall fitting error was less than 10% for $\alpha$ and 5% for $\beta$. Figure 10a,b show the retrieved profiles of, $V_t$, $R_{eff}$, and CRI. The range of EAE suggests that the aerosol layer contained mainly fine mode particles, which is reflected in $R_{eff}$ in Figure 10a,b. The profile of $V_t$ reveals the particles were concentrated mainly below 6500 m. The real part of CRI, $n$, varied between 1.51 and 1.60 while, the imaginary part, $k$, between 0.012 and 0.015, which are in accordance with previous remote or in situ measurements of transported biomass burning aerosols [49,66–68].

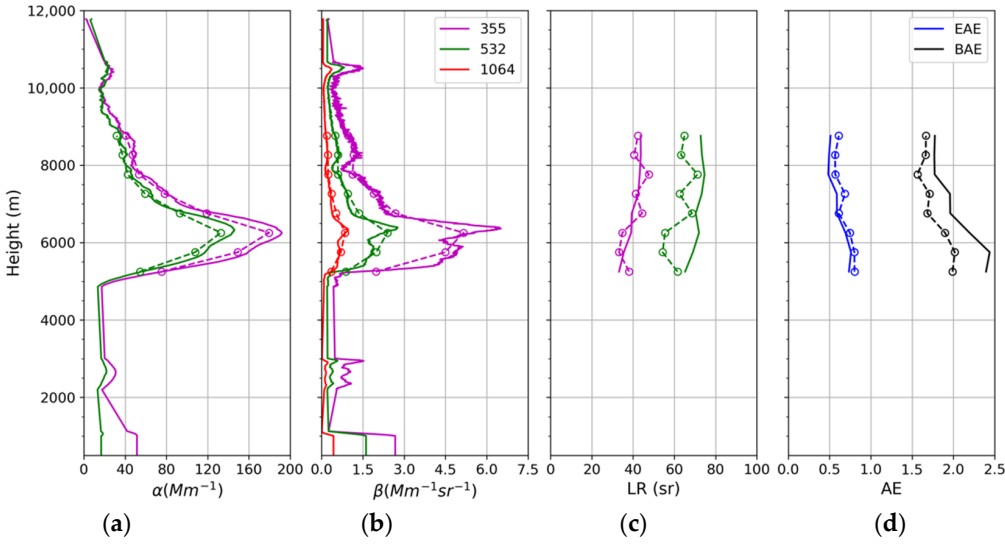

**Figure 9.** Same as Figure 6 but for Case 2: 22:30–03:00 UTC, 11–12 September 2020, Lille. (**a**) $\alpha$; (**b**) $\beta$; (**c**) LR, and (**d**) $AE_{355–532}$. The layer 5000–9000 m was selected and resampled for the retrieval.

Figure 10c shows the retrieved VSDs at 5250 m, 6255 m, and 8265 m, together with the AERONET level 2.0 retrieval on 11 September at 13:55 UTC. It can be seen that the selected layer contains mostly fine mode particles which are well consistent with the fine mode retrieved by AERONET. However, AERONET shows an extra coarse mode accounting for approximately 30% of the total volume concentration. To determine possible reasons for such a difference, note that the columnar $EAE_{340–500}$ measured by AERONET was 0.8 [66], while the $EAE_{355–532}$ of the selected layer measured by LILAS was 0.6. The decrease of EAE could be due to an increase of particle size (i.e., there should be a coarse mode in that layer) or an increase of the imaginary part ($k$) of the CRI when the fine-mode fraction predominates [69]. The later could be reasonable in this case because the AERONET retrieval returned a value of ~0.002, much lower than that retrieved by BOREAL. As mentioned in Section 3.1, here we use $(k_a, \varepsilon_{k_a}) = (0.015, 0.01)$ as the a priori constraint on $k$ because we inferred, with the help of fluorescent measurements of LILAS, that absorbing BBA is concentrated in this layer. However, we also found during the sensitivity study that backscattering kernels corresponding to the coarse-mode region decrease with the increase of $k$, which means $\beta_\lambda$ is less sensitive to the coarse-mode particles, resulting in suppression of the coarse mode under measurement noise. Another possibility results from potential uncertainty of the AERONET retrieval since it is the level 1.5 product (level 2.0 retrieval is unavailable). Therefore, the comparison here is only qualitative.

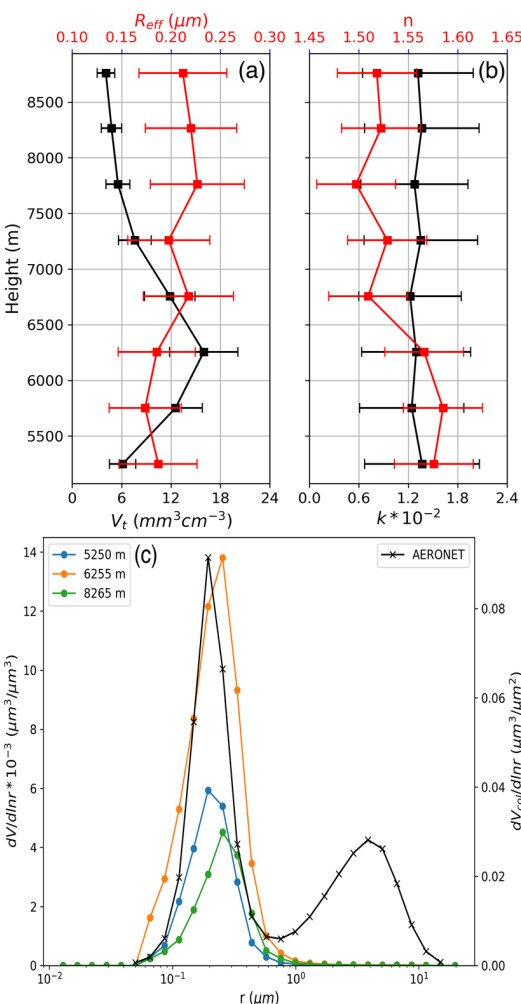

**Figure 10.** Retrievals for Case 2. (**a**) Profiles of $V_{\mathrm{t}}$ and $R_{\mathrm{eff}}$; (**b**) profiles of $n$ and $k$; and (**c**) comparison of layer-resolved VSDs from the LILAS/BOREAL retrieval and column-integral VSD from the AERONET retrieval at 13:55 UTC, 11 September 2020. The error bars in (**a**,**b**) are extracted from Table 3.

*4.3. Case 3: 30–31 May 2020, Lille*

On the night of 30–31 May in the period of 21:00–02:00 UTC, a mixture of pollen grains and urban aerosols from 500 m to 2500 m was observed by LILAS [70]. Considering the wavelength limit of the $3\beta + 2\alpha$ measurements, we retrieved the layer between 1300 m and 2200 m where background urban aerosols mainly concentrated, according to the aerosol classification based on depolarization and fluorescence observations [47]. Spherical and non-absorbing particle assumptions are used in the retrieval.

Figure 11 shows the LILAS measurements and the recalculated measurements in that period. The total processing time was 24 s. Compared with the measurements in previous two cases, the stable and low signals in this case suggest background aerosols, which could consist of fine-mode particles according to the $EAE_{355-532}$. Figure 12a,b show the retrieved profiles of $V_{\mathrm{t}}$, $R_{\mathrm{eff}}$, and CRI. The $R_{\mathrm{eff}}$ varied between 0.12 μm and 0.15 μm, which explains the range of EAE shown in Figure 11. The real part of CRI decreased from 1.57 to 1.50 with the increase of altitude, while the imaginary part of CRI varied between 0.0042 and 0.0049, slightly lower than the a priori value 0.005. Figure 12c shows the retrieved VSDs at different heights, together with the AERONET level 2.0 retrieval on 30 May at 16:28 UTC. The VSDs from the LILAS/BOREAL retrieval are predominated by fine particles with $0.1\,\mu m < r_{\mathrm{v}} < 0.2\,\mu m$, which are well consistent with the fine mode from the AERONET retrieval. The predominated coarse-mode retrieved by AERONET was

very likely to be pollen grains because: (1) the daily cycle of pollen grains, where maximum emission occurs near noon and less emission happens during the night, was validated by in situ measurements [70]; (2) the selected layer excluded the influence of pollen grains according to the aerosol classification result [47]; and (3) the $EAE_{355-532}$ measured by LILAS (~2) was larger than the $EAE_{340-500}$ measured by AERONET (~1.5), and the low value of the imaginary part was retrieved, which indicate the lack of coarse-mode particles in the selected layer.

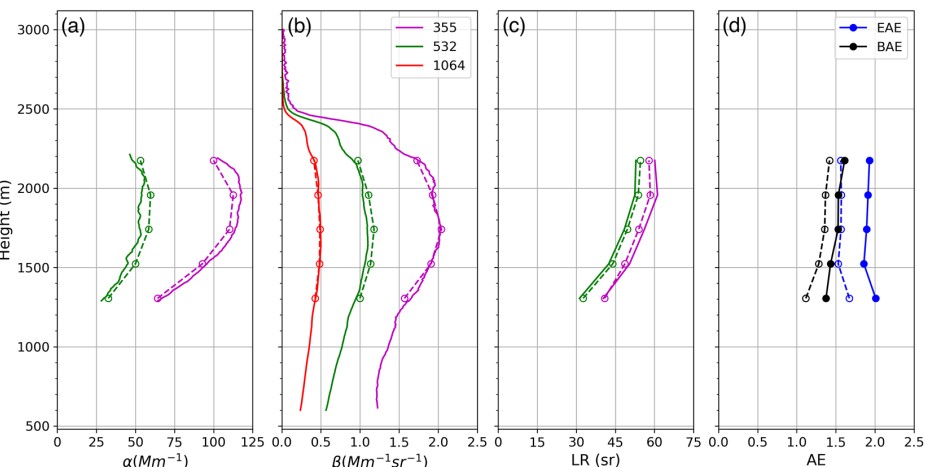

**Figure 11.** Same as Figure 6 but for Case 3: 21:00–03:00 UTC, 11–12 September 2020, Lille. (**a**) $\alpha$; (**b**) $\beta$; (**c**) LR, and (**d**) $AE_{355-532}$. The layer 1300–2200 m was selected and resampled for the retrieval.

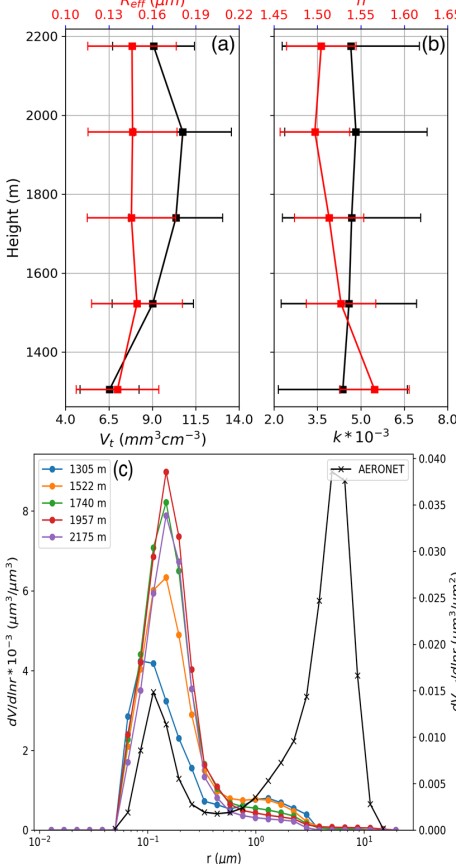

**Figure 12.** Retrievals for Case 3. (**a**) Profiles of $V_t$ and $R_{eff}$; (**b**) profiles of $n$ and $k$; and (**c**) comparison of layer-resolved VSDs from the LILAS/BOREAL retrieval and column-integral VSD from the AERONET retrieval at 16:28 UTC, 30 May 2020. The error bars in (**a**,**b**) are extracted from Table 3.

## 5. Conclusions

The retrieval of height-resolved aerosol microphysical properties is of ever-increasing interest in the field of aerosol remote sensing with the development of lidar networks based on high-performance Mie–Raman lidar systems. In this study, we developed BOREAL, a non-linear inversion algorithm based on maximum likelihood estimation (MLE) to retrieve particle VSD, $V_t$, $R_{eff}$, and CRI ($n - ik$) from the $3\beta$ (backscattering coefficient at 355 nm, 532 nm, and 1064 nm) + $2\alpha$ (extinction coefficient at 355 nm and 532 nm) measured by the Mie–Raman lidar. Compared with other linear retrieval algorithms such as the regularization method and principal component analysis method, BOREAL simultaneously retrieves VSD and CRI by performing optimal searching rather than walking through the whole searching domain, which is evidently more efficient. Based on statistical principles, it is seen that measurement errors are well considered, and their magnitudes serve as scaling factors for the corresponding measurements. At the same time, a priori constraints are treated as virtual measurements with straightforward statistical meaning. Furthermore, the general form of the algorithm will remain unchanged, and computational burden will not evidently increase if more measurements with non-linear forward models are incorporated. To realize stable and realistic retrieval from the ill-posed inversion system, BOREAL (1) utilizes the smoothing constraint on VSD and the a priori constraint on CRI, (2) sets up stopping conditions on the basis of statistical properties, and (3) selects qualified individual solutions derived from a series inversion windows.

We used synthetic optical data ($3\beta + 2\alpha$) generated by different aerosol models to test the performance of BOREAL when one set of a priori constraint on the real part and two sets of a priori constraints on the imaginary part of the CRI were employed. Sensitivity tests show the robustness of the algorithm. For monomodal and bimodal aerosols with $n$ varying in 1.4–1.6 and $k$ varying in 0.005–0.02, VSD, $V_t$, $R_{eff}$, and CRI could be retrieved with acceptable accuracies when measurement uncertainty in each channel is up to 10%. We conclude that $3\beta + 2\alpha$ measurements have limit sensitivity to very low imaginary parts ($k{\sim}0.001$) and large particles, which, in turn, increases retrieval uncertainty for these parameters. At the same time, insufficient information content of $3\beta + 2\alpha$ measurements on the imaginary part increases the influence of the a priori constraint.

We proposed and evaluated an error propagation model, aiming to provide rigorous and real-time estimate of the retrieval covariance matrix, which is a function of the measurement covariance matrix. However, simulation results show that measurement errors in $3\beta + 2\alpha$ data are too large to obey a linear propagation rule, which makes the error propagation model not applicable enough for most cases.

We applied BOREAL to several representative aerosol events: Saharan dust, transported smoke, and background urban aerosols during a pollen season detected by LILAS. The retrieval of the dust case shows good consistency with the result presented by [64], except for overestimates in $V_t$ and $R_{eff}$, which we attribute to the differences in algorithmic principles. The comparisons with AERONET illustrate the advantages and limits of lidar and sun photometer measurements and demonstrate that the aerosol events could be well interpreted by our retrievals.

The next step will also focus on improving the retrieval of CRI, especially the imaginary part. This might be accomplished by further constraining CRI with aerosol-typing results using lidar measurements (for example, see Veselovskii et al. [47]). Another perspective is to incorporate spectral depolarization measurements into the inversion scheme to realize accurate retrieval of non-spherical particles. For this purpose, application of scattering models accurately describing the backscattering of non-spherical particles and assessment of information content of depolarization measurements are needed. At the same time, the first version of BOREAL is being implemented into the AUSTRAL (Automated Server for the Treatment of Atmospheric Lidars) [71] processing and inversion framework to more efficiently evaluate the code with real lidar data and, finally, to implement automated aerosol retrieval and further services.

**Author Contributions:** Conceptualization, Y.C., Q.H. and P.G.; methodology, Y.C., Q.H. and P.G.; software, Y.C.; validation, Y.C.; formal analysis, Y.C., Q.H. and P.G.; investigation, Y.C.; resources, Q.H., P.G., I.V. and T.P.; data curation, Y.C.; writing—original draft preparation, Y.C.; writing—review and editing, Y.C., Q.H., P.G., I.V. and T.P.; visualization, Y.C.; supervision, P.G.; project administration, P.G.; funding acquisition, P.G. All authors have read and agreed to the published version of the manuscript.

**Funding:** This research was funded by French National Research Agency (ANR) through the PIA (Programme d'Investissement d'Avenir) under contract "ANR-11-LABX-0005-01" and by the Regional Council "Hauts-de-France" and the "European Funds for Regional Economic Development" (FEDER).

**Data Availability Statement:** Database of the spherical–spheroidal scattering model can be found in the GRASP-OPEN server https://www.grasp-open.com/products/spheroid-package-release (accessed on 4 December 2022), AERONET data can be found in the AERONET NASA server https://aeronet.gsfc.nasa.gov (accessed on 4 December 2022), and the LILAS measurements and retrievals presented in this study are available on request from the corresponding author.

**Acknowledgments:** The authors acknowledge the support of ESA/QA4EO program, CNRS-INSU National Observation Service PHOTONS-AERONET/EARLINET teams, and Lille University for the observation activity at LOA. The authors thank the PIs of the relevant sites of AERONET for establishing and maintaining the sites used in this investigation. The authors also thank the GRASP team for providing the database of the particle scattering model.

**Conflicts of Interest:** The authors declare no conflict of interest.

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
