# Peer review of "Retrieval of Aerosol Microphysical Properties from Multi-Wavelength Mie–Raman Lidar Using Maximum Likelihood Estimation: Algorithm, Performance, and Application"

_remotesensing, doi:10.3390/rs14246208_

Round 1
Reviewer 1 Report
It is very important how to accurately derive aerosol microphysical properties from the limited lidar measurements. This manuscript described a new algorithm of BOREAL based on the MLE. Some sensitivity test and case studies were also shown.
General comments:
Compared with many other lidar retrieval algorithms (e.g., linear regularization method, principal component analysis, et al.), what are the significant improvements of the BOREAL proposed in this study? Please give a detailed description in the main text so that readers can understand the advanced nature of the algorithm.
Specific comments:
1. Line 50, ‘measurements. [9-11].’ ïƒ ‘measurements [9-11].’
2. Line 72, the format of reference [28] should be revised as Chemyakin et al. [28]. Please check all references in this manuscript.
3. Line 109, should the r1 be equal to rmin, not rmax?
4. Line 112, what is the reason for N set to 8?
5. Line 268, ‘mono-find mode’ should be ‘mono-fine mode’.
6. The coarse mode of AERONET cannot be compared with the retrieval of LILAS/BOREAL in Figure 10c. It is inferred as the contribution of coarse mode particles in the boundary layer by authors. Can authors show some indirect proofs, such as the comparison of column-integrated AOD or AE? Is it possible that the BOREAL algorithm lacks good inversion capability for coarse modal particles? Because the similar situation also occur in Figure 8b and 12c.
Reviewer 2 Report
1. This manuscript describes an algorithm for retrieval of aerosol microphysical properties with Lidar. Also, the comparison and analysis between the error-free and the error-contaminated data have been presented. The manuscript is well organized, and the novelty can meet the requirement of the research paper. Therefore, I recommend publishing this manuscript in Remote Sensing after the minor revision.
2. Is there a comparison of the processing time between the BOREAL and the other algorithms, e.g., the algorithms in reference [64]? It is also a variable that needs to be considered for real-time estimation.
3. In Fig. 7 and 8, there have significant differences between BOREAL and the method in reference [64]. Could the authors explain which one is more accurate? And why?
